# Efficacy of Bacteriophage Cocktail to Control *E. coli* O157:H7 Contamination on Baby Spinach Leaves in the Presence or Absence of Organic Load

**DOI:** 10.3390/microorganisms9030544

**Published:** 2021-03-06

**Authors:** Badrinath Vengarai Jagannathan, Steven Kitchens, Paul Priyesh Vijayakumar, Stuart Price, Melissa Morgan

**Affiliations:** 1Department of Animal and Food Sciences, University of Kentucky, Lexington, KY 40456, USA; badrivj@uky.edu (B.V.J.); paul.v@uky.edu (P.P.V.); 2Department of Pathobiology, College of Veterinary Medicine, Auburn University, Auburn, AL 36849, USA; srk0002@auburn.edu (S.K.); pricesb@auburn.edu (S.P.)

**Keywords:** bacteriophage, *E. coli* O157:H7, organic load, produce, sanitation, dunk tank

## Abstract

Fruits and vegetables are high in nutrients that are essential for a healthy lifestyle. However, they also harbor an extensive array of microorganisms such as bacteria, which can be beneficial, neutral, or pathogenic. Foodborne pathogens can contaminate produce at any stage from the farm to the consumer’s table. Appropriate washing techniques using sanitizers can reduce the risk of pathogen contamination. Issues related to maintaining concentration, efficacy, and other problems have been a challenge for the food industry and, when left unresolved, have led to different outbreaks of foodborne illnesses. In this study, the efficacy of a lytic bacteriophage cocktail was examined for its ability to infect and reduce the contamination of *Escherichia coli* O157:H7 (*E. coli* O157:H7), in media with a high organic load, using a microplate technique. The study was conducted for 3 h to determine if the bacteriophage cocktail could reduce the pathogen in the presence of a high organic load. A significant (*p* < 0.05) reduction in the population of *E. coli* O157:H7 was observed, representing a 99.99% pathogen reduction at the end of 3 h. Fresh spinach leaves were washed in sterile potable or organic water (~9000 ppm organic load) containing *E. coli* O157:H7 and a bacteriophage cocktail to study the effectiveness of bacteriophages against the foodborne pathogen. Results indicated that the bacteriophage significantly (*p* < 0.05) reduced the contamination of *E. coli* O157:H7 in both situations. The study also demonstrated the bacteriophages’ ability to infect and reduce the pathogen in an organic-rich environment. This characteristic differs from commercially available sanitizers that have demonstrated a tendency to bind with the available organic load. Thus, these studies highlight the advantage of employing bacteriophages during produce wash to eliminate foodborne pathogen contamination on fruits and vegetables.

## 1. Introduction

Consumption of fresh fruits and vegetables is often encouraged by government agencies in many countries as an essential part of a balanced diet and healthy lifestyle [1]. Fresh fruits and vegetables consumed raw, such as leafy greens, are recognized to be potential vehicles for human pathogens that are traditionally associated with foods of animal origin [1]. Developed and developing countries are strengthening current food safety systems worldwide to face real and perceived food safety challenges encountered by their food industries [2]. Each year, *Escherichia coli* O157:H7 (*E. coli* O157:H7) causes 73,000 illnesses in the United States, resulting in an estimated 2168 hospitalizations and 61 deaths [3]. Infections from *E. coli* O157:H7 are often associated with the consumption of meat or meat products. However, several outbreaks have been traced back to the consumption of contaminated produce, such as radishes and pre-packaged spinach [1]. The first outbreak associated with *E. coli* O157:H7 in produce was reported in 1991 [4]. Since then, raw produce has been viewed as a potential vehicle for pathogens that cause various foodborne illnesses. Decontaminating fruits, vegetables, and meat products has always been considered a challenge in the food industry [5]. The most common ways of limiting microbial growth on fruits and vegetables are washing them with water or rinsing them with a solution containing antimicrobials such as chlorine-based chemicals [5]. Washing produce is considered a vital aspect of postharvest processing that has a significant influence on maintaining product quality and safety [6]; therefore, wash-water quality is one of the most critical parameters and plays a crucial role in reducing contamination during postharvest washing, cooling, and sanitizing operations [7]. Although water is a useful tool in reducing contamination, it can also aid in pathogen transfer through cross-contamination during postharvest activities as improperly treated wash-water can serve as a vector for the contamination of fresh produce [8,9]. It is well known that produce industries, especially those that handle fresh-cut produce, rely on wash-water quality and sanitizers to minimize microbial counts and achieve an extended shelf-life for their products [9]. The food industry has long used chlorine-based sanitizers to maintain the safety of its products [10]. However, recent outbreaks associated with produce have raised concern about traditional sanitizer efficacy in ensuring product safety. Chlorine-based sanitizers utilized today only give 1–2 log reduction of the pathogens [11,12]. Thus, current investigations seek chlorine-based sanitizer alternatives that can provide safety to the products without compromising quality and shelf life [10]. Bacteriophages (commonly called phages) are bacterial viruses that selectively infect bacteria and disrupt their metabolism, resulting in lysis of the host bacterial cell [13]. Since phages are highly specific, they can target a specific pathogen without harming any beneficial microorganisms [14]. Phages have been proven to act as a natural antimicrobial to fight against bacterial infections in humans, animals, and crops [15]. Several studies have focused on phages as a promising alternative that can be used in the food industry to eliminate bacterial contamination, especially on produce [11,16,17,18]. The focus of this research was to investigate the ability of a bacteriophage cocktail to act as a natural intervention to lyse *E. coli* O157:H7 on spinach leaves during a simulated dunk-tank wash in the presence and absence of an organic load.

## 2. Materials and Methods

### 2.1. Bacterial Culture for Microplate and Produce Wash Study

Pathogenic *Escherichia coli* O157:H7 (*E. coli* O157:H7) (ATCC 35150) was obtained from a freezer stock. Working stock cultures were prepared by re-suspending cells into tryptic soy broth (TSB) (Difco, Becton-Dickenson Labs, Sparks, MD, USA) and incubating for 48 h at 37 °C before streaking the cultures on MacConkey Agar (MAC) (Difco, Becton-Dickenson Labs, Sparks, MD, USA) and Sorbitol MacConkey Agar (SMAC) (Difco, Becton-Dickenson Labs, Sparks, MD, USA) for isolation. After incubation for 24 h at 37 °C, the characteristics of the colonies were observed and individual colonies were picked from SMAC and transferred into TSB tubes (supplemented with 5 mM of magnesium sulfate (MgSO_4_) (Fisher Scientific, Fair Lawn, NJ, USA) and calcium chloride (CaCl_2_) (Fisher Scientific, Fair Lawn, NJ, USA)) using a sterile technique. Cultures were grown on tryptic soy agar (TSA) (Difco, Becton-Dickenson Labs, Sparks, MD, USA) for 24 h at 37 °C and then stored at refrigeration temperature, 4 °C, until needed for propagation.

### 2.2. Bacteriophage Cocktail Preparation

Four bacteriophages (C14 s, V9, L1, and LL15), specific to *E. coli* O157:H7 were obtained from bovine feces. The dairy herd bacteriophages were isolated and characterized by the Auburn University College of Veterinary Medicine. Bacteriophages were grown for 24 h at 37 °C with host *E. coli*. Phages were then separated via centrifugation at 28,500 relative centrifugal force (RCF) for 20 min in the presence of chloroform. The phages were then filter-sterilized through a 0.22µ filter (Merck Millipore Ltd., Tullagreen, Carrigtwohill, Ireland) into working stock containers. A bacteriophage titer was determined to confirm phage activity. The host strain for all the bacteriophages was *E. coli* O157:H7 (ATCC 35150). The phage titer was approximately 10^9^ plaque-forming units (PFU)/mL for the phage cocktail [19]. Equal numbers of individual bacteriophage types were mixed in a sterile test tube, and the required volume was pipetted to make the phage cocktail right before every experiment.

### 2.3. Turbidometric Growth Inhibition Assays in the Presence of Organic Load

An equal volume (1 mL) of C14 s, L1, LL15, and V9 phages were mixed in a sterile tube to obtain a phage cocktail. Sterile Dey-Engley neutralizing buffer (DE) broth (Difco, Becton-Dickenson, Sparks, MD, USA) and DE broth with 100 µL of *E. coli* O157:H7 (ATCC 35150) were used as control treatments. DE broth with 100 µL phage cocktail acted as a negative control to show that the bacteriophages do not contribute to turbidity at 660 nm. One hundred microliters of *E. coli* O157:H7 (ATCC 35150) (~1.00 × 10^9^ CFU/mL) was inoculated into DE broth and distributed to wells in a 96-well flat-bottom microtiter plate (Thermo Fisher Scientific). One hundred microliters of bacteriophage cocktail was added to the wells and mixed by aspiration using a multi-channel micro-pipette, which contributed to a multiplicity of infection (MOI) of 1. Using a microplate reader (BioTek, Synergy 4, Winooski, VT, USA), the turbidity analysis settings were developed from a previously determined procedure [19]. The settings for the turbidity analysis were as follows: temperature of 37 °C (range: 36.5–37 °C), number of flashes as 1, measurement mode as absorbance, measurement wavelength of 660 nm, start kinetic (run: 3:00:00, interval 00:30:00), shake duration (orbital) of 10 s (s), shake intensity as medium, total measurement time of 24 h, and unit as optical density (OD). A lid was used to prevent evaporation of the liquid and well-to-well contamination of the 96-well plate. The OD660 values were plotted against time to illustrate the phage cocktail preparations’ antimicrobial activity against *E. coli* O157:H7. Samples from the microplate wells were collected at the end of 3 h for both the control and treatment. These samples were then diluted using sterile peptone water (1:10) and plated (100 µL) on premade TSA plates (supplemented with 5 mM CaCl_2_ and MgSO_4_) and incubated overnight at 37 °C. The control and treatment samples were plated in duplicates, and the entire experiment was repeated three times.

### 2.4. Initial Produce Rinse to Reduce Background Microbial Contamination on Spinach Leaves

Fresh baby spinach leaves were purchased from a local grocery chain. Spinach leaves were transferred into a sterile filter bag (Fisher brand–blender bags) and treated with a 2% lactic acid solution (Fisher Scientific) for 15 min followed by 100-ppm bleach water (Clorox, Oakland, CA, USA) for an additional 15 min. Leaves were then set under UV light for 15 min to reduce the background population of microorganisms and dissipate any residual chlorine present on the leaves (Figure 1).

### 2.5. Wash Solution for the Simulated Dunk Tank

Twenty milliliters of double-distilled deionized sterile water was used for the initial experiment to study the efficacy of the bacteriophage cocktail against *E. coli* O157:H7 in the absence of an organic load (Figure 1). For the following study, 20 mL of sterilized DE broth containing approximately 9810 ppm of dissolved organic matter (casein, 1660 ppm; yeast extract, 830 ppm; dextrose, 3330 ppm; Tween, 80–1660 ppm; and lecithin, 2330 ppm) was used as a wash solution to determine the ability of the bacteriophage cocktail to infect *E. coli* O157:H7 in the presence of an organic load. Control samples were treated similarly with organic-load wash-water without the bacteriophage cocktail. In both studies, the samples were immersed in the wash solution for the full contact time of 10 min.

### 2.6. Application of Sterile Potable Wash-Water Solution Containing E. coli O157:H7 and Bacteriophage Cocktail in a Simulated Dunk Tank

Fresh spinach leaves, after the initial produce rinse step, were separated into three different treatments: negative control (NC), positive control (PC), and bacteriophage cocktail treatment (BCT) (Figure 1). The NC was washed spinach without any other treatment to validate the initial wash’s efficacy and to observe if any background microorganisms were still present on the leaves. The PC sample had leaves that were dunk-washed for 10 min in 20 mL sterile potable water containing 1500 µL of *E. coli* O157:H7 (~1.0 × 10^9^ CFU/mL). The BCT sample had leaves dunk-washed in 20 mL sterile potable water with a combination of 1500 µL of *E. coli* O157:H7 (~1.0 × 10^9^ CFU/mL) and 3000 µL of bacteriophage cocktail (MOI: 2.3). The samples were placed in a sterile sampling bag at room temperature and sampled at 0, 3, 6, 9, and 12 h.

### 2.7. Application of Sterile Wash Solution Containing 9810 ppm of Organic Load Comprising E. coli O157:H7 and Bacteriophage Cocktail in a Simulated Dunk Tank

A similar procedure from the above study was applied with DE broth instead of the sterile potable water to mimic an organic load present in the wash water. The positive control and the bacteriophage cocktail treatment for this experiment were represented as O-PC and O-BCT, respectively. All the samples were packed in a sterile sampling bag and sampled at 0 and 3 h.

### 2.8. Recovery of Bacteria

Produce was rinsed with 1 mL of sterile phosphate buffer. Samples were held at room temperature and massaged through the sampling bag for one minute. Serial dilutions of the sample rinse were made in phosphate buffer (pH 7.4–7.5). The dilutions were then plated on premade TSA plates supplemented with 5 mM MgSO_4_ and 5 mM CaCl_2_.

### 2.9. Statistical Analysis

The data were analyzed using the GLIMMIX procedure in SAS 9.4. A linear mixed model was used where the response variable was expressed in Log10 values, and the fixed effects were treatment, time, and the interaction between treatment and time. A random intercept for the subject defined with a specific treatment within the study was included in the model. The *p*-values were adjusted for multiple comparisons using the Tukey–Kramer method.

## 3. Results

### 3.1. Microplate Growth Inhibition Assay and Plate Count Study of Bacteriophage Cocktail against E. coli O157:H7 in the Presence of Organic Load

Positive controls of *E. coli* O157:H7 demonstrated a typical logarithmic growth pattern over the test period. The bacteriophage cocktail showed significant inhibition of the pathogen (Figure 2). The bacteriophage cocktail preparation decreased the growth of *E. coli* O157:H7 (*p* < 0.01) in a controlled environment in the presence of a 9810-ppm organic load from 1.00 × 10^9^ CFU/mL to 1.84 × 10^3^ CFU/mL in the treatment resulting in a 99.99% reduction of the pathogen. The study demonstrated that phages are highly specific to the host pathogen despite being in a relatively concentrated organic load. The phages specifically targeted the bacteria and infected and reduced the host population, in contrast with commercially used sanitizers such as bleach, which are less effective in the presence of an organic load. Chlorine has a higher affinity toward organic matter, which depletes its effectiveness against microorganisms.

### 3.2. Effect of Bacteriophage on Potable Water Wash Solution Containing E. coli O157:H7 Inoculated Spinach in a Simulated Dunk Tank

The initial produce rinse successfully inhibited the growth of background flora on fresh spinach. The plate count (<1.00 log CFU/mL) on the negative control (NC) indicated that the initial rinse effectively rinsed the background microflora. Table 1 shows the bacteriophage cocktail’s efficacy in reducing *E*. *coli* O157:H7 on spinach washed in potable water containing the phage cocktail compared with the control wash. The 10-min contact time in the bacteriophage-containing wash solution resulted in a significant reduction (*p* < 0.05) of the pathogen at the end of 3 h compared to the positive control (PC). A gradual recovery of the pathogen numbers occurred in the bacteriophage cocktail treatment (BCT) samples after 3 h until 12 h. The statistical analysis indicated that despite the recovery, the BCT was still significantly different from the PC. Therefore, the phage disinfectant treatment (BCT) was significantly effective (*p* < 0.05) in reducing the population of *E. coli* O157:H7 on the spinach leaves.

### 3.3. Effect of Sterile Potable Wash Solution Containing 9810 ppm of Organic Load Comprising E. coli O157:H7 and Bacteriophage Cocktail in a Simulated Dunk Tank

The initial produce rinse was once again influential in reducing the spinach’s background microflora (<1.00 log CFU/mL). Table 2 shows the bacteriophage cocktail’s efficacy in reducing *E*. *coli* O157:H7 on spinach washed in the organic water (9810 ppm organic load) containing the phage cocktail compared with the control wash. The 10-min contact time for the wash solution resulted in a significant reduction (*p* < 0.01) of 99.99% of the pathogen at the end of 3 h compared to the O-PC. This study also illustrated the bacteriophage’s specificity and its ability to effectively reduce *E. coli* O157:H7 despite being in an environment with a high organic load.

Comparison between the growth of *E. coli* O157:H7 in the positive controls of the potable wash (Table 1) and organic wash-water treatments (Table 2) at the end of 3 h demonstrated an increase in the pathogen population on the leaves that were washed using the organic-rich water (O-PC). The presence of organic materials in the wash water and the 3 h hold might have contributed to this increase in population. Regardless of the increase in the pathogen population, a reduction in the population of the pathogen (99.99%) can be seen in the bacteriophage treatment (O-BCT).

## 4. Discussion

The postharvest wash procedure is considered to be a critical control point (CCP) for the removal of any field-assimilated contamination in the fresh-produce industry [20]. Chlorine is one of the most commonly used sanitizers in the produce industry. The internationally recommended concentration for chlorine-based compounds used for rinsing produce is between 50 and 100 ppm of free chlorine [21]. This range is reported to achieve a pathogen reduction of approximately 1–2 log CFU/g [22]. The effectiveness of chlorine-based sanitizers decreases in the presence of organic matter in produce wash-water [23]. Thus, pre-treatment removal of organic matter and continuous monitoring of sanitizer concentration are suggested for the practical use of sanitizers in the food industry [23,24]. Despite these efforts, bacterial outbreaks in the fresh-produce industry continue to occur.

As an antimicrobial, bacteriophage has proven to be efficient in reducing the population of *E. coli* O157:H7 in foods. Previous studies have evaluated the effectiveness of bacteriophage cocktails specific to different pathogens such as *Salmonella*, *E. coli* O157:H7, and *Listeria*. Leverentz et al. [25] observed a reduction in the *Salmonella enteritidis* population on fresh-cut honeydew melon after spot-treating the infected portion with a bacteriophage cocktail. The pathogen population was reduced 3.5 and 2.5 log CFU/wound after the treated melons were stored at 5–10 and 20 °C, respectively [25]. Similarly, treating fresh-cut honeydew melons with listeriophages (spray or aliquots) reduced the population of *L. monocytogenes* by 2 to 4.6 log units compared to the untreated controls when stored at 10 °C [17]. Most of these studies either spot- or spray-treated the samples with bacteriophage to demonstrate their effectiveness against the pathogen.

Although the previous studies demonstrated the efficacy of using bacteriophages against pathogens, they did not apply the results to real-time scenarios. For that reason, this study sought to determine the effectiveness of bacteriophages in dunk tanks, a commonly used wash procedure. Dunk tanks, also referred to as immersion or dip tanks, carry a significantly higher risk of cross-contamination of pathogens between contaminated and clean produce [24]. Immersion-washer procedures employ dumping, submerging, or floating of the produce in wash-water with or without sanitizer [6]. The potential for pathogen uptake by produce through infiltration is a significant concern for the food industries that use dunk tanks or other immersion techniques [6]. Pathogen infiltration can occur through the stem scar, calyx, or other surface openings naturally present on fresh produce. Apart from this, if the washing procedure is not monitored or managed correctly, it can cause produce injury, cross-contamination, or internalization of the pathogen [6]. For instance, from 2000 to 2002, the United States faced a multistate outbreak of *Salmonella* serotype Ponna associated with consumption of cantaloupes imported from Mexico. An on-farm investigation of the outbreak revealed that the melons were washed and cooled in contaminated wash-water, which could have been the possible contamination source [6,26]. A multistate outbreak of *Salmonella enterica* serotype Newport, associated with mango consumption in the United States, led to 78 confirmed cases of salmonellosis in 13 states. Penteado et al. in 2004 investigated the recall by recreating the washing scenarios to study the pathogen’s ability to contaminate the fruit during the washing process. The team tested *Salmonella’s* ability to internalize in fresh mangoes during a simulated postharvest insect disinfection procedure. Pathogen internalization was observed when heat-disinfected mangoes were cooled using the contaminated water. The study concluded that low wash-water quality and improper chlorination could have served as a vector for contamination of the mangoes [27]. These outbreaks emphasize the need for a practical technique during production and postharvest activities to mitigate the risk of pathogen contamination on fresh produce. Employing commercial chlorine-based sanitizers alone has not solved pathogen contamination since only 1–2 log CFU reduction is expected under specified conditions [11,12]. Employing bacteriophage as a disinfectant effectively reduces the population of *E. coli* O157:H7 in fresh produce without the use of chemical sanitizers. Abuladze et al. 2008 studied a bacteriophage cocktail’s ability to reduce *E. coli* O157:H7 contamination on broccoli, spinach, tomato, and ground beef. Treatment with the bacteriophage cocktail resulted in a significant reduction (*p* ≤ 0.05) of the pathogen with minimal recovery as incubation time increased. The percent reduction on broccoli was 99.5%, 99%, and 97%; tomatoes, 99%, 94%, and 96%; and spinach, 100%, 99.6%, and 91%, at 24, 120, and 168 h, respectively. Data obtained in the current study were similar, wherein the bacteriophage cocktail in sterile potable wash-water reduced the *E. coli* O157:H7 population by 99.77% at the end of 3 h compared to the control. In the case of wash-water containing a high organic load, the bacteriophages contributed to a 99.99% reduction at the end of 3 h. The sterile potable wash-water study indicated a recovery of the pathogen as incubation time increased. The emergence of phage-resistant bacterial mutants, transduction of undesirable characteristics among bacteria, and environmental conditions have been suggested as problems that can potentially reduce the effectiveness of a phage treatment [28]. However, several studies have suggested that employing a cocktail of different bacteriophages could reduce the likelihood of generating a mutant [29,30]. One possible explanation could be the mechanism of phage attachment. Phages tend to attach to different receptors found on the host bacteria, and the mutation of one specific phage receptor would not alter the attachment site for another phage [30]. Additionally, as a result of all the genetic changes that the pathogen endures developing resistance to the phage cocktail, it is improbable that the resulting pathogen would be as virulent or survive due to the fitness trade-off that it undergoes internally, which includes colonization defects, reduced virulence, and resensitization to antibiotics [31]. Because phages are ubiquitous, the isolation of new phages specific to the pathogen that exhibits a difference in the attachment mechanism can be used to update phage cocktails and make them effective against the development of phage-resistant mutant strains. Although the use of bacteriophages requires that investigators exercise caution to ensure their specificity against the target pathogen, the bacteriophage cocktails used in this study have been previously investigated for their specificity and also have demonstrated infectivity only for *E. coli* O157:H7 strains [19].

Numerous foodborne outbreaks of *E. coli* O157:H7 have been caused by <20.00 CFU/g or even <1.00 CFU/g of the pathogen [32]. However, in a real-life scenario, a very high load of *E. coli* O157:H7 contamination on produce is very unlikely to occur [5]. The amount of *E. coli* O157:H7 used in this experiment was several thousand-fold higher than that associated with an outbreak. These levels were utilized to study and visualize the efficacy of the bacteriophage cocktail. Several studies reported by other investigators concluded that a lower phage-host ratio could be better at reducing the pathogen [5,33]. Therefore, increasing the phage concentration might help achieve a more significant reduction of the pathogen during produce wash. Environmental conditions can also play a significant role in a phage’s viability; thus, studying the effect of these conditions will help to ensure the continuous performance of the phage against the pathogen. Phages employed for this study have also been shown to be resistant to 100-ppm chlorine and 100-ppm SaniDate 5.0 for up to 3 h [19]. Thus, developing a multilevel sanitation system that employs both a sanitizer and bacteriophage combination might be one of the solutions to reduce pathogen contamination on fresh produce.

## 5. Conclusions

In summary, the results from this study indicate that the bacteriophages can be effectively used as a tool in reducing *E. coli* O157:H7 contamination on fresh produce. Bacteriophages irrespective of being subjected to a complex environment (organic load), significantly reduced the population of the pathogen. Future studies involving combination treatment methods or hurdle technology in large-scale trials might be required to verify this possibility and help mitigate the exposure of fresh produce to foodborne pathogens.

## Figures and Tables

**Figure 1 microorganisms-09-00544-f001:**
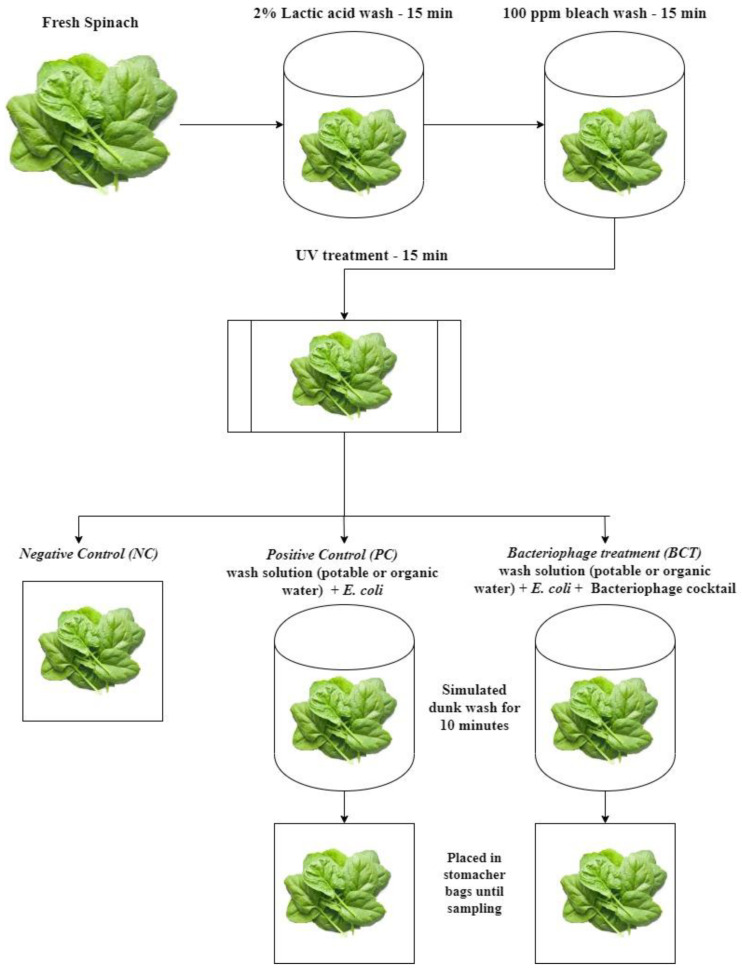
Schematic flow of initial produce rinse and dunk wash of spinach leaves.

**Figure 2 microorganisms-09-00544-f002:**
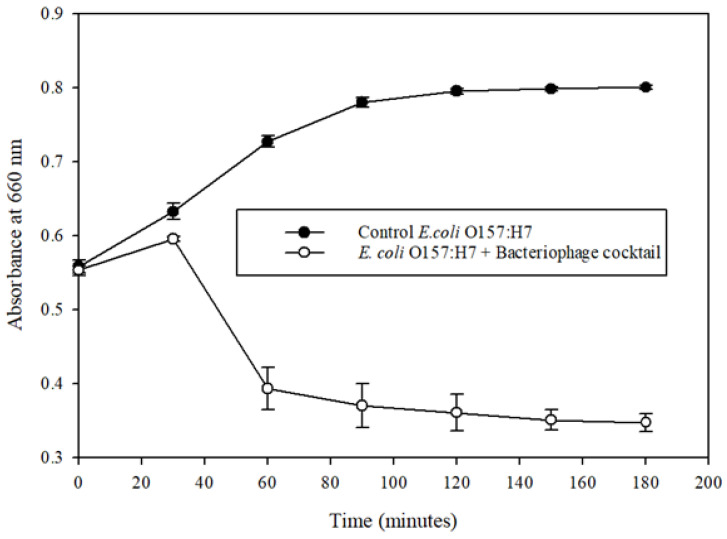
Microplate growth inhibition assay shows a bacteriophage cocktail against *Escherichia coli* O157:H7 (*E. coli* O157:H7) (ATCC 35150) in the presence of organic load. The data points represent the means of triplicate replication, and the error bars represent the standard deviations of three independent experiments. The bacteriophage cocktail reduced the population of *E. coli* O157:H7 (ATCC 35150) significantly (*p* < 0.05) compared to the control.

**Table 1 microorganisms-09-00544-t001:** Reduction of *E. coli* O157:H7 (ATCC 35150) on spinach via postharvest pathogen control measures when using bacteriophage cocktail wash solution made with potable water in a simulated dunk tank. The data and standard deviation (SD) represent the mean of triplicate replication.

Wash Treatment	Wash Time (min)	Sampling Time (h)	*E. coli* O157:H7 Population (log CFU/mL (SD))	*E. coli* O157:H7 Log Reduction (Log CFU/mL)	Percentage Reduction (%) of *E. coli* O157:H7
Negative Control (NC)	-	0	<1.00		
		3	<1.00		
		6	<1.00	-	-
		9	<1.00		
		12	<1.00		
Positive Control (PC)	10	0	6.25 (0.15)		
		3	6.46 (0.22)		
		6	7.11 (0.10)	-	-
		9	7.35 (0.04)		
		12	7.39 (0.07)		
Produce wash + Bacteriophage cocktail (BCT)	10	0	5.97 (0.53)	0.28	47.05
		3	3.80 (0.17)	2.66	99.77
		6	5.02 (0.42)	2.09	99.18
		9	5.21 (0.20)	2.14	99.27
		12	5.45 (0.36)	1.94	98.85

**Table 2 microorganisms-09-00544-t002:** Reduction of *E. coli* O157:H7 (ATCC 35150) on spinach via postharvest pathogen control measures when using bacteriophage cocktail wash solution made with water containing 9810 ppm of organic load in a simulated dunk tank. The data and standard deviation (SD) represent the mean of triplicate replication.

Wash Treatment	Wash Time (min)	Sampling Time (h)	*E. coli* O157:H7 Population (Log CFU/mL (SD))	*E. coli* O157:H7 Log Reduction (Log CFU/mL)	Percentage Reduction (%) of *E. coli* O157:H7
Negative Control (NC)	-	0	<1.00	-	-
		3	<1.00		
Positive Control (O-PC)	10	0	6.46 (0.06)	-	-
		3	7.06 (0.11)		
Produce wash + Bacteriophage cocktail (O-BCT)	10	0	6.15 (0.09)	0.31	56.89
		3	2.94 (0.06)	4.12	99.99

## Data Availability

Data is contained within the article.

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
