# Peer review of "Efficacy of Bacteriophage Cocktail to Control E. coli O157:H7 Contamination on Baby Spinach Leaves in the Presence or Absence of Organic Load"

_microorganisms, 2021, doi:10.3390/microorganisms9030544_

Round 1

Reviewer 1 Report

This is an interesting manuscript on the efficacy of bacteriophage to control E. coli O157:H7 on baby spinach. The manuscript contains interesting data.  However, some points should be improved:  

Results.

Lines 191-200 Although the authors describe the treatments applied in material and methods and Table 1, I suggest to detail the treatment: negative control (NC) (line 192). positive control wash (line 196), phage disinfectant treatment (line 197). Including the description, it is easier to follow the manuscript.

Lines 191-192 Could the authors indicate the initial microbial counts before rinsing? Or only E coli O157:H7 counts were followed?. 

Lines 195-196 The authors stated “The 10 -minute… compared to the PCT”. The treatment applied should be specified. Is it the wash solution containing the bacteriophage?. 

Tables 1 and 2. It should be included a column indicating the log reduction of E. coli O157:H7, and not only the percentage of reduction (%).  E. coli O157:H7 counts media and standard deviation should be given. Number of replicates should be indicated in the Tables. Did the authors find significant differences between the same treatment and different sampling times? A between the different treatments at the same sampling time. These information should be shown in the Tables.

Author Response

Lines 191-200 Although the authors describe the treatments applied in material and methods and Table 1, I suggest to detail the treatment: negative control (NC) (line 192). positive control wash (line 196), phage disinfectant treatment (line 197). Including the description, it is easier to follow the manuscript.

Response: The acronyms NC, PC, and BCT on lines 192,196, and 197 were detailed out as suggested.

Lines 191-192 Could the authors indicate the initial microbial counts before rinsing? Or only E coli O157:H7 counts were followed?. 

Response: Initial counts were not particularly analyzed and only E. coli O157:H7 counts were followed. But, it was found that unrinsed samples had contamination with Pseudomonas and the initial rinse took care of reducing the contamination which was in turn checked by plating the negative control for all sample times.

Lines 195-196 The authors stated “The 10 -minute… compared to the PCT”. The treatment applied should be specified. Is it the wash solution containing the bacteriophage?. 

Response: Yes, the treatment name is added to lines 195-196.

Tables 1 and 2. It should be included a column indicating the log reduction of E. coli O157:H7, and not only the percentage of reduction (%).  E. coli O157:H7 counts media and standard deviation should be given. Number of replicates should be indicated in the Tables. Did the authors find significant differences between the same treatment and different sampling times? A between the different treatments at the same sampling time. These information should be shown in the Tables.

Response: Log reduction added to tables 1 and 2, along with the number of replicates and Standard deviation.

Reviewer 2 Report

A well-designed experiment. I enjoyed the simplicity and relevance of the paper. There are some significant editorial suggestions, but they can be addressed.
